# Circulating Exosomes Inhibit B Cell Proliferation and Activity

**DOI:** 10.3390/cancers12082110

**Published:** 2020-07-29

**Authors:** Jan C. Schroeder, Lisa Puntigam, Linda Hofmann, Sandra S. Jeske, Inga J. Beccard, Johannes Doescher, Simon Laban, Thomas K. Hoffmann, Cornelia Brunner, Marie-Nicole Theodoraki, Patrick J. Schuler

**Affiliations:** Department of Otorhinolaryngology and Head and Neck Surgery, Ulm University, 89075 Ulm, Germany; jan.schroeder@uni-ulm.de (J.C.S.); lisa.puntigam@uni-ulm.de (L.P.); Linda.Hofmann@uniklinik-ulm.de (L.H.); sandrajeske001@gmail.com (S.S.J.); inga.beccard@uni-ulm.de (I.J.B.); Johannes.Doescher@uniklinik-ulm.de (J.D.); simon.laban@uniklinik-ulm.de (S.L.); T.hoffmann@uniklinik-ulm.de (T.K.H.); Cornelia.Brunner@uniklinik-ulm.de (C.B.); Marie-Nicole.Theodoraki@uniklinik-ulm.de (M.-N.T.)

**Keywords:** exosomes, B cells, Head and Neck Cancer, Bruton’s tyrosine kinase

## Abstract

(1) Background: Head and neck squamous cell carcinoma (HNSCC) is characterized by a distinctive suppression of the anti-tumor immunity, both locally in the tumor microenvironment (TME) and the periphery. Tumor-derived exosomes mediate this immune suppression by directly suppressing T effector function and by inducing differentiation of regulatory T cells. However, little is known about the effects of exosomes on B cells. (2) Methods: Peripheral B cells from healthy donors and HNSCC patients were isolated and checkpoint receptor expression was analyzed by flow cytometry. Circulating exosomes were isolated from the plasma of HNSCC patients (*n* = 21) and healthy individuals (*n* = 10) by mini size-exclusion chromatography. B cells from healthy individuals were co-cultured with isolated exosomes for up to 4 days. Proliferation, viability, surface expression of checkpoint receptors, and intracellular signaling were analyzed in B cells by flow cytometry. (3) Results: Expression of the checkpoint receptors PD-1 and LAG3 was increased on B cells from HNSCC patients. The protein concentration of circulating exosomes was increased in HNSCC patients as compared to healthy donors. Both exosomes from healthy individuals and HNSCC patients inhibited B cell proliferation and survival, in vitro. Surface expression of inhibitory and stimulatory checkpoint receptors on B cells was modulated in co-culture with exosomes. In addition, an inhibitory effect of exosomes on B cell receptor (BCR) signaling was demonstrated in B cells. (4) Conclusions: Plasma-derived exosomes show inhibitory effects on the function of healthy B cells. Interestingly, these inhibitory effects are similar between exosomes from healthy individuals and HNSCC patients, suggesting a physiological B cell inhibitory role of circulating exosomes.

## 1. Introduction

With the recent advent of effective cancer immunotherapy, the interaction between tumor cells and the immune system has gained a lot of attention. Tumors inhibit the anti-tumor immune response [1], and tumor entities differ with regard to their immune-suppressive properties. Head and Neck Squamous Cell Carcinoma (HNSCC) is classified as one of the most immune-suppressive cancers, making it a pivotal candidate for immunotherapy [2].

In HNSCC, several mechanisms contribute to tumor immune escape: (I) downregulation of antigen expression, (II) upregulation of immune-suppressive mediators such as PD-L1, and (III) education of host cells [3]. In addition, the tumor microenvironment (TME) of HNSCC is highly inflammatory. High concentrations of inhibitory cytokines and metabolites, such as adenosine, promote tumor progression and suppress anti-tumor immunity [2,3,4,5]. While tumor-infiltrating effector cells such as T, B, NK, and dendritic cells are functionally impaired in HNSCC [6,7], regulatory T cells (Treg) are actively recruited and contribute to the immune escape [8].

Among the diverse mechanisms that tumors employ to impair anti-tumor immunity, tumor-derived exosomes (TEX) have recently emerged as an additional important player [9,10]. Exosomes are extracellular vesicles that are physiologically produced by all cells across several species and can be isolated from all body fluids [11,12]. They resemble their mother cells in RNA, DNA, and protein content and their composition of surface molecules enables them to deliver a combination of signals between cells across long distances [13]. Tumors exploit these features by producing high amounts of TEX, which promote tumor growth and progression and mediate tumor immune escape by reprogramming target immune cells into compliance [14].

Intriguingly, HNSCC-derived exosomes mediate both immune suppression and education of host cells, such as inhibition of CD4^+^ T cells, increased apoptosis of CD8^+^ T cells, and induction of Treg [15]. Moreover, both the amount of exosomes in the plasma of HNSCC patients and the level of PD-L1 on the surface of exosomes correlate with disease stage and prognosis [16].

The effects of exosomes on a wide variety of leukocytes and other cells in the TME have been investigated, but only little is known about the interaction of tumor-derived exosomes and B cells. The role of B cells in anti-tumor immunity is controversial, since both pro- and antitumorigenic effects have been suggested [17]. This apparent contradiction is possibly explained by the diverse effects of different B cell subpopulations. In particular, regulatory B cells (Breg) are correlated with tumor progression and inhibit anti-tumor immune response [17]. Regarding the effects of TEX on B cells, studies of hepatocellular carcinoma have shown that Breg are induced in the TME by factors, which could not be further identified [18]. Other groups demonstrated that Breg could be induced by exosomes purified from hepatocellular carcinoma cell culture supernatants or blood plasma of esophageal cancer patients [19,20]. To our knowledge, there is no evidence so far on the effect of HNSCC-derived TEX on B cells.

In this study, we compared the effects of exosomes purified from the plasma of healthy individuals and HNSCC patients on B cells.

## 2. Results

### 2.1. Clinicopathological Characteristics of HNSCC Patients

Clinicopathological data for all patients enrolled in this study are shown in Table 1. The patients were predominantly male and the average age was 64 years (exosome isolation) and 59 years (B cell isolation). Anatomical locations of primary tumors were the pharynx, the larynx, and the oral cavity. Exosomes were isolated from 21 patients, 15 (71%) of which presented with high primary tumor stage (T3-T4). B cells were isolated from 23 patients, 11 (47.8%) of which presented with high primary tumor stage. Most patients were nodal-positive but none had distant metastases. Three out of twenty-one (14%) patients that enrolled for exosome isolation were HPV-positive. Four out of twenty-one patients (19%) had been treated with radiochemotherapy at the time of enrolment.

### 2.2. Expression of Checkpoint Receptors on B Cells

To compare the expression of checkpoint receptors between B cells from healthy individuals and HNSCC patients, B cells were isolated and analyzed by flow cytometry. The expression of PD-1 and LAG3 was significantly increased in B cells isolated from HNSCC patients (Figure 1, *p* ≤ 0.05).

### 2.3. Characterization of Plasma-Derived Exosomes

Extracellular vesicles isolated from plasma were characterized by TEM, Western blot and nanoparticle tracking. Vesicular morphology, negative contrasting, and a diameter between 30 and 150 nm were evident in TEM images (Figure 2A). The expression of the specific exosomal markers TSG101, CD9 and CD63 was demonstrated by Western blot, while exosomes did not contain the negative markers ApoA1 or Grp94 in large quantities (Figure 2B). Size range measured by nanoparticle tracking confirmed diameters between 30 and 150 nm (Figure 2C). The average concentration of plasma-derived exosomes was 77.1 µg/mL (HNSCC) and 58.8 µg/mL (NC) (Figure 2D).

### 2.4. Exosomes Inhibit Proliferation and Survival of B Cells In Vitro

To investigate biological effects of plasma-derived exosomes on healthy human B cells, a B cell exosome co-culture protocol was established. B cells were isolated from healthy individuals and co-cultured with plasma-derived exosomes obtained from NC or HNSCC for up to 4 days. After 24 h, B cells were stimulated by adding CD40L and IL-4 to ensure survival and to induce activation and proliferation. Approximately 48 h after stimulation, B cells started to proliferate, which was evident as a formation of cell colonies under the microscope (Figure 2E). This effect was not seen in B cells co-cultured with exosomes derived from NC or HNSCC.

A flow chart for the sequence of experiments in the co-culture protocol is displayed in Figure 3A. Expression of CD19 on B cells was confirmed by flow cytometry (Figure 3B,C). B cell viability was assessed by trypan blue stain after 2–4 days and the amount of surviving B cells in culture was determined by the number of measured B cells in flow cytometry (Figure 3D,E). Co-culture of B cells with exosomes derived from NC or HNSCC reduced B cell viability and B cell count after 2 and 4 days. Overall, plasma-derived exosomes from both HNSCC patients and healthy individuals inhibited colony formation and survival of B cells.

### 2.5. Exosomes Modulate the Expression of Checkpoint Receptors on B Cells

We hypothesized that exosome-induced changes in proliferation and viability were accompanied by changes in checkpoint receptor expression on B cells. Surface expression of co-inhibitory and co-stimulatory checkpoint receptors was measured during co-culture on days 2, 3, and 4 by flow cytometry (Appendix A). Due to increasing variance after more than 2 days, we focused on changes observed on day 2 of co-culture. Exosomes derived from NC, but not from HNSCC, induced an increase in the expression of PD1, CTLA4, LAG3, and CD137 as compared to PBS-controls (Figure 4). The expression of BTLA was reduced by both exosomes from NC and HNSCC. The expression of GITR was only reduced by HNSCC-derived exosomes. No significant changes were observed for the expression of TIM3, CD27, and OX40. Control groups included unstimulated B cells and stimulated B cells, which were cultured without exosomes (PBS). Expression levels are shown as mean fluorescence intensity (MFI). There were no relevant differences between exosomes from post-radiochemotherapy patients and exosomes from other HNSCC patients.

### 2.6. Effects of Exosomes on CD39/CD73 Expression

Changes in the frequency of CD39^+^CD73^+^ Breg were measured by flow cytometry after co-culture with plasma-derived exosomes (Figure 5A) and no significant differences were observed. In contrast, MFI of CD39 was significantly (*p* = 0.0059) reduced after co-culture with NC or HNSCC exosomes for 2 days (Figure 5B). To evaluate the functional consequences of this reduction, ATP hydrolysis by B cells was measured via luciferase assay, which showed a small, non-significant reduction in ATP hydrolysis by B cells after co-culture with HNSCC or NC exosomes (Appendix A).

### 2.7. Exosomes Reduce the Activity of the B-Cell Receptor Pathway

Several of the checkpoints affected by exosome treatment are known to affect or regulate BCR signaling. As a readout for activity of the BCR pathway, we assessed phosphorylation of Bruton’s Tyrosine Kinase (BTK) by flow cytometry. Stimulation of the BCR pathway by anti-µ-F(ab)_2_ antibody fragments results in increased phosphorylation of BTK (Appendix A). Co-culture of B cells with either NC or HNSCC-derived exosomes for up to 16 h prior to stimulation, induced a time-dependent decrease in p-BTK expression (Figure 6).

## 3. Discussion

Most of the work on TEX and their impact on the anti-tumor immune response has focused on T cells. Here, we investigate the effects of plasma-derived exosomes from NC and HNSCC patients on B cells. Our data suggest that these exosomes inhibit the proliferation, viability, and function of B cells, in vitro.

Before investigating the effects of exosomes on healthy B cells, we looked for differences in checkpoint receptor expression between B cells isolated from healthy individuals and HNSCC patients. We found an increase in the frequency of PD-1^+^ and LAG3^+^ B cells in the plasma of HNSCC patients as compared to healthy individuals (Figure 1), which parallels findings from studies of hepatocellular and thyroid cancer patients [18,21,22]. Except for BTLA and CD27, the expression of all analyzed checkpoint receptors was low. There was a slight trend towards higher expression but also higher variance in B cells derived from HNSCC patients.

We have previously shown that both exosomes from healthy individuals and exosomes from HNSCC patients carry immunosuppressive cargo, such as PD-L1, FasL, and CTLA4 [10,15]. This is in line with our data showing suppressive effects of both NC and HNSCC exosomes on B cells.

Exosomes isolated from plasma represent mixtures of exosomes derived from various cells, such as hematopoietic cells, mesenchymal cells, endothelium, and organ-specific cells, and in the case of malignancy, TEX [13,23,24,25]. Interestingly, we have shown that in HNSCC, T cell-derived CD3^+^ exosomes constitute around 50% of all plasma-derived exosomes [23], as opposed to a frequency of 30% in healthy individuals. Moreover, CD3^+^ exosomes carry similar immune-suppressive mediators as do CD3^neg^ exosomes although differences in functionality were described [23,26]. This is in line with the known role of exosomes in immune regulation and signaling between immune cells [27]. Endogenous exosomes have immune-suppressive properties, for example in an antigen-specific fashion after oral tolerance induction [28]. Furthermore, exosomes from T cells are known to induce apoptosis via delivery of death ligands [29]. Also, immune regulatory exosomes from Treg, natural killer cells, neutrophils, and endothelial cells have been reported [29].

It is assumed that the amount and the composition of exosomes in the serum is very variable between the individual patients and changes can occur in between days. In order to minimize additional variability, the blood from HNSCC patients was always drawn on the day of surgery. In the case of NC exosomes, all individuals with a history of malignancy, autoimmune disorders or a current state of inflammation were excluded. Age and gender distribution between NC and HNSCC patients enrolled for exosome isolation were significantly different with NC patients being younger and with a higher female fraction. However, both types of exosomes inhibited proliferation and viability of B cells in a similar way. Therefore, we do not expect to see a difference when enrolling older NC patients. Still, differences seen in checkpoint receptor expression after co-cultivation of B cells with HNSCC versus NC exosomes might be confounded by the different demographic composition of HNSCC and NC patients.

In contrast to our findings, Mao et al. reported no changes in B cell proliferation after co-culture with plasma-derived exosomes from healthy individuals [19]. These differences might be due to different experimental conditions. In the reported experiments, B cells were stimulated with both CD40L and anti-IgM antibody and cultured at a higher density (5 × 10^5^ cells/well vs. 2.5 × 10^4^ cells/well in our study). Although exosomes were added at higher quantities (10 or 20 µg/well), the exosome—B cell ratio was much lower. Additionally, proliferation was measured after 48 h compared to 2–4 days in our study. These effects may account for the missing exosome-induced inhibition. Furthermore, exosomes were purified using a polymer-based precipitation solution. It is known that different methods of exosome isolation can produce diverging results and that polymer-based precipitation can result in higher protein and RNA yield, but also higher contamination as compared to miniSEC [30].

Immune checkpoints such as PD-1 and CTLA4 are a family of immune regulatory receptors, which were initially characterized in T cells and crucially affect survival, phenotype, and activity, thereby directing immune responses [31]. Several of these checkpoints are known to affect B cell activity, especially in the TME.

Flow cytometry revealed various changes in checkpoint receptor expression on B cells after co-culture with plasma-derived exosomes (Figure 4). Increases were seen in PD-1, CTLA4, LAG3, and CD137 expression after co-incubation with NC exosomes, but not HNSCC exosomes. PD-1 expression on activated B cells has been associated with decreased BCR signaling and upregulation of regulatory properties [32,33]. Subsets of PD-1^high^ Breg have been described in hepatocellular carcinoma and thyroid cancer patients [18,21,22]. Mao et al. described no alteration of PD-1 levels on Breg after treatment with NC exosomes [19]. The possible reasons for this discrepancy were addressed above.

CTLA4 expression in B cells can be induced by activated T cells [34] and partially inhibits immunoglobulin production [35]. High levels of CTLA4 were reported in B cell malignancies but also tumor-associated B cells in malignant melanoma [36,37]. LAG3, an inhibitory checkpoint receptor, and CD137, which promotes B cell proliferation [38], can also be induced by activated T cells [38,39,40]. Co-culture with both NC and HNSCC exosomes reduced the expression of inhibitory receptor BTLA and stimulatory receptor GITR. BTLA is a negative regulator of BCR signaling that is downregulated after B cell activation [32,41]. Furthermore, BTLA^+^ CD19^hi^ B cells were associated with poor outcome in ovarian cancer [42]. GITR expression in human B cells has not been reported before, but studies in mice revealed a weak expression, which was increased by activation of BCR and CD40 and suppressed by cytokines secreted from T cells [43,44].

In general, exosomes can affect target cells both by surface interaction or by fusion with the cell membrane and uptake of contents. From our studies with T cells, we know that the former mechanism is likely responsible for exosomal effects on these lymphocytes, while antigen-presenting cells such as dendritic cells actively take up exosomes [45]. In the case of B cells, it was specifically shown by our group that after co-incubation CD19+ B cells can internalize exosomes [46].

In addition to changes in checkpoint receptors, we observed a reduction in CD19 expression after 3 to 4 days of co-culture with HNSCC exosomes (Appendix A). CD19 is a pan B cell marker acting as a co-stimulatory receptor for both BCR-dependent and -independent signaling [47]. The reduced expression might reflect an inhibitory effect of exosomes but can also be due to changes of B cell maturation in vitro.

Some changes were only observed after treatment with NC, but not HNSCC exosomes, suggesting that these are part of physiological exosomal functions. It is important to note that there was higher inter-individual heterogeneity in experiments with HNSCC exosomes compared to NC. Increased PD-1, CTLA, LAG3, and CD137 levels were seen in only some experiments with HNSCC exosomes, whereas exosomes from other patients did not show any effect. These findings are in line with the high heterogeneity of HNSCC. Some of our findings mimic changes seen after B-T cell interaction (CTLA4, LAG3, CD137, GITR) or activation of B cells in general (PD-1, BTLA, GITR). However, the overall effect here is not activation, but inhibition of B cell survival and activity, possibly due to the distinct combination of effectors in exosomes.

A large proportion of mature B cells express CD39 and CD73 on their surface [48]. CD39 and CD73 are ectonucleotidases that convert extracellular ATP via adenosine 5′-monophosphate to adenosine (ADO). We have previously shown, that the expression of CD39 is directly correlated to the production of exogenous adenosine by a variety of immune cells [48]. In addition, our experiments have shown that exogenous adenosine is highly immunosuppressive. Adenosine is hydrolyzed from exogenous ATP. Therefore, especially in the tumor microenvironment, the adenosine pathway can have a major influence on the immune system as a high concentration of the substrate ATP is set free by dying cells [5,49].

In this study, the expression of CD39, but not CD73, was reduced after co-culture with HNSCC or NC exosomes (Figure 5). CD39 was initially described as an activation marker and we have previously shown its upregulation after activation with CD40L and IL-4 [48]. Due to its role in ADO generation, the downregulation of CD39 on B cells may result in a decreased immune suppression due to a lower concentration of adenosine. On the other hand, CD39 is a B cell activation marker and downregulation may be due to an interference of exosomes with B cell activation by our stimulation protocol.

B cells did not proliferate and were less viable after treatment with plasma-derived exosomes. Furthermore, changes in the expression of several proteins, which affect BCR signaling and decreased BCR activity, were observed (Figure 6). The BCR pathway controls B cell survival, proliferation, and activation and is a key-player for signaling machinery of B cells [50]. Exosomes might inhibit B cell proliferation and survival by engaging the expression level of co-receptors that negatively regulate BCR signaling. Further studies are needed to establish causality and investigate functional consequences beyond proliferation, such as cytokine production, antibody generation, and regulatory properties towards T cells. Furthermore, the complex interaction between B cells and exosomes in the context of HNSCC and the TME needs to be addressed by performing in vivo studies and investigating the effects of circulating exosomes on B cells derived from HNSCC patients.

## 4. Materials and Methods

### 4.1. Patients

Peripheral blood specimens were collected from healthy individuals or HNSCC patients seen at the ENT Department of Ulm University Hospital between 2014 and 2019. The study was approved by the local ethics committee (#255/14). Clinicopathological data of HNSCC patients and corresponding controls are detailed in Table 1. Healthy individuals who donated blood for B cell isolation to be used in B cell co-cultivation with exosomes (*n* = 70) were 60% male and had an average age of 31 years (range: 17–80 years).

### 4.2. Exosome Isolation by Mini Size-Exclusion Chromatography (Minisec)

MiniSEC was used to isolate plasma-derived exosomes and was routinely performed as detailed in our previous publications [15,51]. In short, plasma samples were centrifuged at 2000 *g* for 10 min, at 10,000 *g* for 30 min, and ultrafiltrated using a 0.22 µm filter to remove impurities. One milliliter of plasma was added to a miniSEC column and eluted with phosphate-buffered saline (PBS). The fourth fraction was collected and concentrated using Amicon Ultra columns (Merck Millipore, Burlington, MA, USA). Extracellular vesicles isolated from plasma were characterized according to the MISEV 2018 guidelines for the definition of extracellular vesicles [52]. Isolation of exosomes was confirmed by assessing particle size and count using nanoparticle tracking (ZetaView PMX-220, Particle Metrix), morphology by transmission electron microscopy, and origin from the endosomal compartment by Western Blot.

### 4.3. Bicinchoninic Acid (BCA) Protein Assay

Exosomal protein content was evaluated using BCA assay (Pierce BCA Protein Assay Kit, Thermo Fisher Scientific, Waltham, MA, USA). Ten microliter aliquots of exosome samples isolated by miniSEC and dissolved in PBS were stained with BCA and incubated for 30 min at 37 °C. An albumin standard curve was always prepared in parallel und used for quantitation. Final concentrations of exosomes were between 200 µg/mL for incubation with B cells and 1.000 µg/mL for electron microscopy.

### 4.4. Transmission Electron Microscopy (TEM)

Exosomal morphology was evaluated using TEM. Exosomes in PBS were layered on copper-grids in chloroform with 0.125% formvar and subsequently stained using uranyl acetate. Images were acquired using a JEOL 1400 electron microscope.

### 4.5. SDS-PAGE and Western Blot

To confirm expression of exosomal markers, exosome samples were analzyed by SDS-PAGE and Western Blot. Electrophoresis was performed using MiniProtein Precast gels (Bio-Rad, Hercules, CA, USA) at a voltage of 120–200 V. Gels were carefully transferred to nitrocellulose membranes and blotted using the TransBlot Turbo System, also by Bio-Rad. The membranes were then washed three times using Tris-buffered saline + 0.05% Tween (TBST) and incubated with primary antibodies at 4 °C overnight, and washed four in TBST and incubated with secondary antibodies for 40 min. Bands were visualized by adding SuperSignalTM West Dura Extended Duration Substrate (Thermo Fisher Scientific). Images were acquired and processed using the ChemiDoc MP Imaging System and ImageLab software (both from Bio-Rad). Uncropped Western Blot images are shown in Appendix A. The following primary antibodies were used: CD9 (10626D), CD63 (10628D), TSG101 (PA5-31260) all from Invitrogen (Carlsbad, CA, USA), EpCAM (MA5-13917, Thermo Fisher Scientific), ApoA1 (3350) and Grp94 (2104), both from Cell Signaling Technology (Danvers, MA, USA).

### 4.6. B Cell Culture

Peripheral blood mononuclear cells were isolated from fresh blood samples donated by healthy individuals (mean age 31 years) using LeukoSep. B cells were isolated by CD19-negative selection using the EasySep Human B Cell Isolation Kit (StemCell, Vancouver, Canada) and immediately used for experiments. B cells were cultured for up to 4 days at 37 °C in RPMI buffer (Thermo Fisher Scientific) containing 10% exosome-depleted fetal bovine serum (also Thermo Fisher Scientific) and 1% ZellShield (Minerva Biolabs, Berlin, Germany) at a concentration of 125,000 cells/mL in 96-well plates. Some B cells were co-cultured with plasma-derived exosomes at a quantity of 7.5 µg per well. After 24 h, B cells were stimulated with 2 µg/mL CD40 Ligand (CD40L, R&D Systems, Minneapolis, MN, USA) and 1500 IU/mL Interleukin 4 (IL-4, CellGenix, Freiburg, Germany). Each day, B cells were observed for cluster formation by light microscopy, and viability was evaluated using Trypan Blue stain.

### 4.7. Flow Cytometry

After 2–4 days, B cells were gently harvested and incubated with fluorochrome-labeled detection antibodies for 30 min at 4 °C in the dark, washed two times, and resuspended in 400 µL washing buffer (PBS containing 0.05% bovine serum albumin). Fluorescence was detected with a Gallios flow cytometer (Beckman Coulter, Brea, CA, USA) and analyzed with Kaluza software from Beckman Coulter. The following antibodies were used for detection: BTLA Brilliant Violet 421 (344512), CD19 APC-Fire 750 (302257), GITR Brilliant Violet 421 (371207), OX40 PE-Cy5 (350009) and TIM-3 Pacific Blue all from Biolegend (San Diego, CA, US), CD137 PE (12-1379-42), CD27 PE-Cy5 (15-0279-42), CD39 PE-Cy7 (25-0399-42), CD73 FITC (11-0739-42), LAG3 PE (12-2239-42) and PD-1 PE (12-2799-41) from eBioscience (San Diego, CA, US), CD86 Alexa Fluor 700 (561124) and CTLA4 PE-Cy5 (561717) from BD Biosciences (Franklin Lakes, NJ, USA).

### 4.8. Intracellular p-BTK Staining

For intracellular phosphostaining of Bruton’s Tyrosine Kinase (p-BTK), B cells were cultured for up to 16 h in RPMI medium containing 10% FBS and 1% ZellShield at a concentration of 100,000 cells/mL in 15 mL falcons. Some B cells were co-cultured with plasma-derived exosomes at a quantity of 10 µg per falcon. The B cell receptor (BCR) was then stimulated for 5 min by adding 5 µg/mL anti-µ-F(ab)_2_ antibody fragment (Affinipure, Jackson ImmunoResearch, West Grove, PA, USA). To enable intracellular staining, B cells were fixed by adding paraformaldehyde to a final concentration of 4% for 15 min and permeabilized by adding PBS containing 0.1% Triton X-100 and incubated at room temperature (RT) in the dark for 30 min. Next, B cells were washed two times in PBS containing 0.05% BSA and stored in ice-cold 1:1 PBS-methanol solution at −20 °C for at least 10 min to stabilize phosphate residues. To detect p-BTK, permeabilized B cells were washed twice and then incubated with anti-p-BTK (Y223) PE antibody (BD Biosciences, Franklin Lakes, NJ, USA) for 45 min at RT in the dark. Flow cytometry was performed as described above.

### 4.9. ATP Hydrolysis Assay

To measure changes in ATP hydrolysis capacity of B cells after co-culture with exosomes, B cells were harvested after 3 days of co-culture and incubated with 20 µM ATP for up to 120 min. ATP concentration was then determined through addition of luciferase and substrate solution (ATPlite luminescence Assay System, PerkinElmer, Waltham, USA) and subsequent bioluminescence measurement.

### 4.10. Statistical Analysis

Statistical analysis was performed using Graph-Pad Prism version 5 (GraphPad Software, La Jolla, CA, USA). Box and whiskers graphs depict the median as a horizontal line, while boxes range from the 25th to 75th percentiles and whiskers extend from the minimal to the maximal data point. Scatter plots show all data points and depict the mean as a horizontal line. Line and bar graphs show mean and standard error of the mean. The Mann Whitney test was used for comparisons between groups. *p*-value < 0.05 was used to evaluate significance of the data.

## 5. Conclusions

Tumor-derived exosomes mediate immune suppression in HNSCC by regulating and educating various immune cells. Here, we demonstrate inhibitory effects of circulating exosomes from both healthy individuals and HNSCC patients on B cells. Exosomes inhibit B cell proliferation and survival and change the surface expression of checkpoint molecules on B cells. Furthermore, B cell receptor signaling was reduced after co-culture with exosomes. These data suggest a physiological B cell inhibitory role of circulating exosomes that may be co-opted by HNSCC, where exosomes are produced in high quantities.

## Figures and Tables

**Figure 1 cancers-12-02110-f001:**
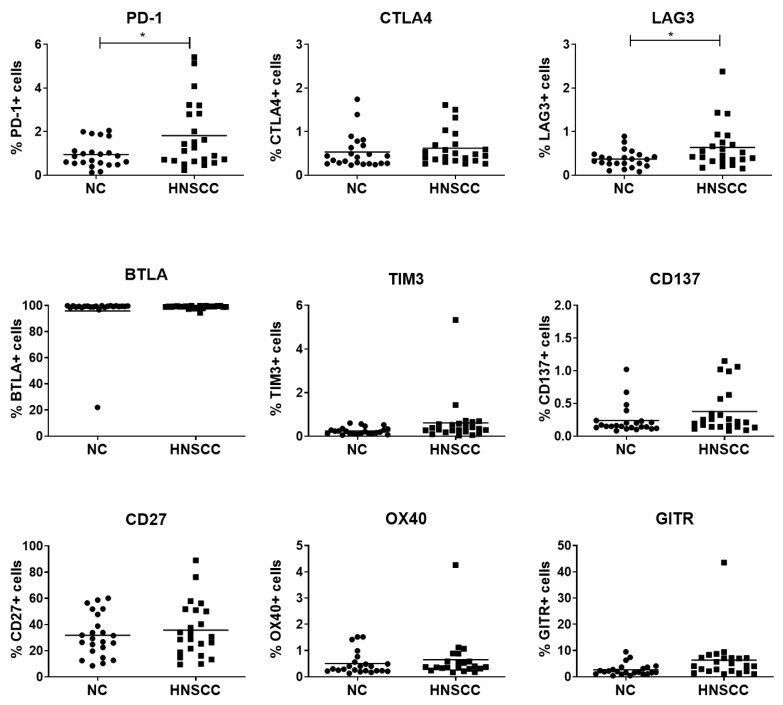
B cells were isolated from healthy individuals and HNSCC patients and analyzed by FACS. Shown is the frequency of cells expressing PD-1, CTLA-4, LAG3, BTLA, TIM3, CD137, CD27, OX40, and GITR. The expression of PD-1 and LAG3 was significantly increased in B cells isolated from HNSCC patients. *n* = 23, each dot represents a B cell sample from a separate individual. *: *p* < 0.05. HNSCC, B cells isolated from blood plasma of HNSCC patients. NC = no cancer, B cells isolated from blood plasma of healthy volunteers.

**Figure 2 cancers-12-02110-f002:**
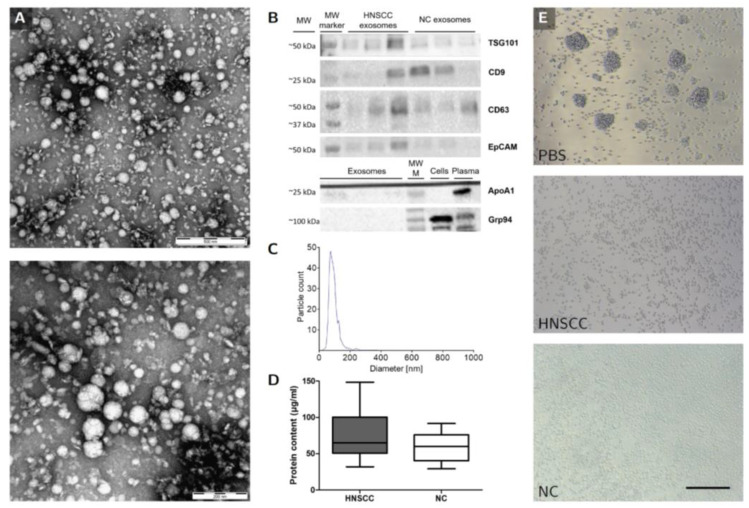
Successful isolation of exosomes from blood plasma was verified by Transmission Electron Microscopy (TEM), Western Blot, and Nanoparticle tracking. (**A**) Two representative TEM graphs showing negatively stained exosomes isolated from an HNSCC patient. As indicated by the size bars, exosomes vary in diameter between 30 and 150 nm and have round to oval shapes. Size bar on the top TEM graph = 500 nm, size bar on the bottom TEM graph = 200 nm. (**B**) Western Blot analysis of exosomes was performed to confirm the expression of exosomal markers TSG101, CD9 and CD63 and the expression of epithelial cell marker EpCAM (upper frame). Exosomes were also analyzed for negative markers ApoA1 and Grp94 along with plasma (diluted 50× in PBS) and cell lysate samples as positive controls. MW marker, positive control molecular weight marker. (**C**) Size distribution of exosomes was measured by nanoparticle tracking. The mean diameter was 86.8 nm. The maximal and minimal diameters were 257.5 nm and 22.5 nm, respectively. The 90th and 10th percentile were at 121.2 and 53.6 nm, respectively. (**D**) Protein content of exosomes was determined by Bicinchoninic Acid (BCA) Assay. Average protein content: 80.9 g/mL (HNSCC exosomes), 69.2 g/mL (healthy volunteer exosomes). *n* = 23 (HNSCC), *n* = 10 (NC). HNSCC, exosomes from blood plasma of HNSCC patients. NC = no cancer, exosomes from blood plasma of healthy volunteers. (**E**) B cells that were not co-cultured with exosomes exhibited colony formation under the light microscope (top frame). This was not observed with B cells co-cultured with HNSCC exosomes or NC exosomes (middle and bottom frame, respectively). Size bar = 200 µm. *n* ≥ 5 (HNSCC: 8, NC: 5, Control: 8).

**Figure 3 cancers-12-02110-f003:**
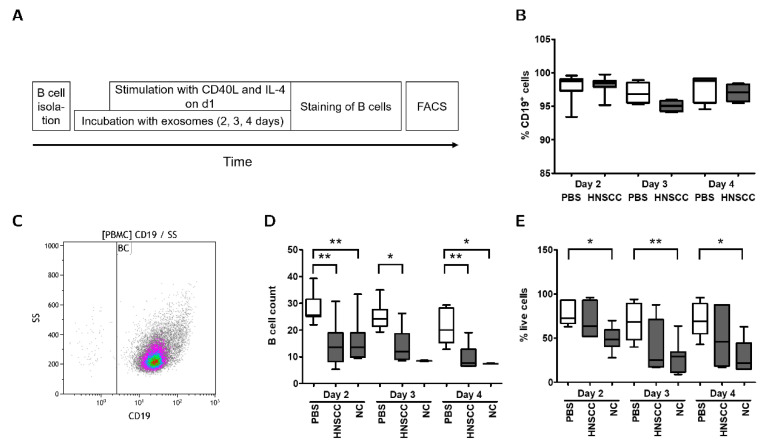
B cells were isolated from citrate blood samples drawn from healthy donors. (**A**) B cells were co-cultured with exosomes for up to 4 days and stimulated with CD40 Ligand (CD40L) and Interleukin 4 (IL-4) on day 1. Each day, B cells were studied by microscopy and counted using trypan-blue stain and an automated cell counter. (**B**) B cell purity as assessed by CD19-positivity of cells was above 90% throughout culture. *n* = 8 (Day 2), *n* = 4 (Days 3 & 4). (**C**) Purity of isolated B cells was confirmed by anti-CD19 stain and subsequent FACS analysis. Purity after isolation was always above 98%. (**D**) B cell survival as measured by FACS cell count. B cell counts were reduced after co-culture with both healthy volunteer and HNSCC exosomes. There were no significant differences between HNSCC and NC exosomes. *n* = 10 (HNSCC exosomes on d2); *n* = 9 (NC exosomes and PBS on d2); *n* = 6 (PBS and HNSCC exosomes on d3 & d4); *n* = 2 (NC exosomes on d3); *n* = 3 (NC exosomes on d4). (**E**) B cell viability as measured by trypan blue staining and automated counting was significantly lower after co-culture with healthy volunteer exosomes. A similar, but non-significant trend was observed after co-culture with HNSCC exosomes. There were no significant differences between HNSCC and NC exosomes. *n* ≥ 5 (HNSCC: 5 [d2], 9 [d3], 6 [d4]; NC: 6 [d2], 9 [d3], 5 [d4]; PBS: 5 [d2], 7 [d3], 6 [d4]). **: *p* < 0.01; *: *p* < 0.05. HNSCC, exosomes from blood plasma of HNSCC patients. NC = no cancer, exosomes from blood plasma of healthy volunteers.

**Figure 4 cancers-12-02110-f004:**
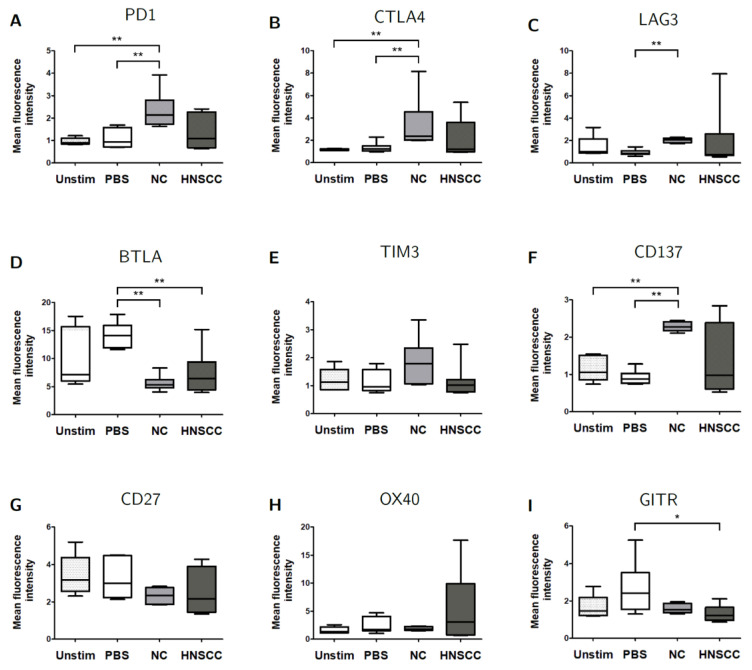
B cells were harvested after 2 days of co-culture with either NC or HNSCC exosomes or PBS and stained for FACS analysis. (**A**) The expression of PD-1 on B cells was increased after co-culture with NC exosomes. (**B**) The expression of CTLA4 on B cells was increased after co-culture with NC exosomes. (**C**) The expression of LAG3 on B cells was increased after co-culture with NC exosomes. (**D**) The expression of BTLA on B cells was reduced after co-culture with NC or HNSCC exosomes. (**E**) Expression of TIM3 on B cells. (**F**) The expression of CD137 on B cells was increased after co-culture with NC exosomes. (**G**) Expression of CD27 on B cells. (**H**) Expression of OX40 on B cells. (**I**) The expression of GITR on B cells was reduced after co-culture with HNSCC exosome. **: *p* < 0.01; *: *p* < 0.05, *n* = 8 (HNSCC), *n* = 6 (NC), *n* = 5 (Unstim). Unstim = Unstimulated B cells, NC = no cancer (exosomes from blood plasma of healthy volunteers), HNSCC, exosomes from blood plasma of HNSCC patients.

**Figure 5 cancers-12-02110-f005:**
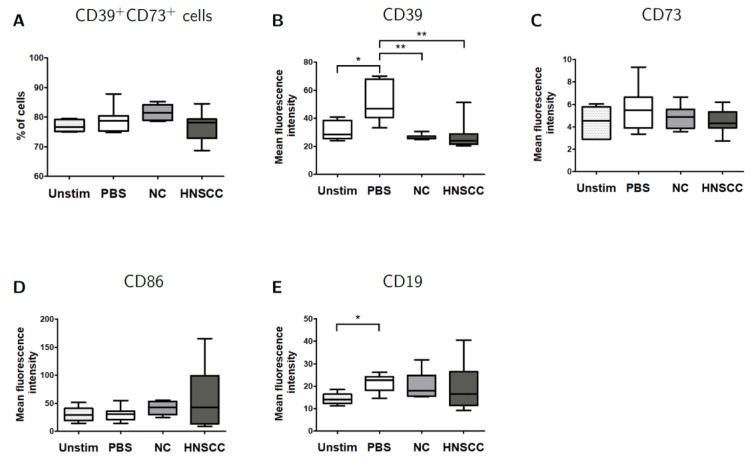
B cells were harvested after 2 days of co-culture with either NC or HNSCC exosomes or PBS and stained for FACS analysis. (**A**) Frequency of CD39^+^CD73^+^ regulatory B cells. (**B**) The expression of CD39 on B cells was reduced after co-culture with NC or HNSCC exosomes. (**C**) Expression of CD73 on B cells. (**D**) Expression of CD86 on B cells. (**E**) The expression of CD19 on B cells was increased by stimulation with CD40L and IL-4. **: *p* < 0.01; *: *p* < 0.05, *n* = 8 (HNSCC), *n* = 6 (NC), *n* = 5 (Unstim). Unstim = Unstimulated B cells, NC = no cancer (exosomes from blood plasma of healthy volunteers), HNSCC, exosomes from blood plasma of HNSCC patients.

**Figure 6 cancers-12-02110-f006:**
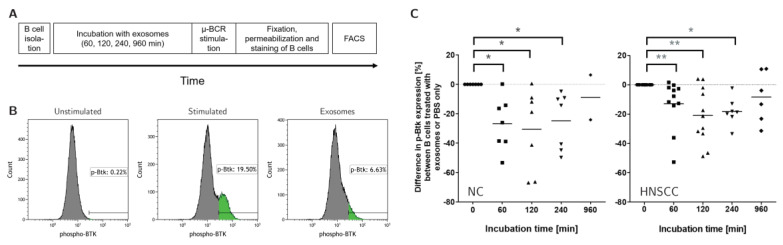
Effects of Plasma-derived Exosomes on the activity of the B-cell-receptor pathway. (**A**) B cells isolated from healthy donors were incubated with exosomes for up to 16 h. After stimulation of the B-cell receptor (BCR), fixation and permeabilization, B cells were stained with an antibody against phosphorylated Bruton’s Tyrosine Kinase (p-BTK) to assess the activity of the BCR pathway. (**B**) Exemplary FACS plots of B cells after BCR stimulation and exosome treatment show that BCR stimulation results in enhanced expression of p-BTK, while exosome treatment reduces this effect. (**C**) Treatment with both NC and HNSCC exosomes reduced the expression of p-BTK after BCR stimulation. B cells were incubated with exosomes or PBS as a control for 60 to 960 min. The percent difference of p-BTK expression after exosome-treatment vs. PBS treatment is depicted on the *y*-axis. *n* = 7 (NC), *n* = 11 (HNSCC), each dot represents a controlled experiment. ** *p* < 0.01, * *p* < 0.05. HNSCC, exosomes from blood plasma of HNSCC patients. NC = no cancer, exosomes from blood plasma of healthy volunteers.

**Table 1 cancers-12-02110-t001:** Clinicopathological parameters for patients with head and neck squamous cell carcinoma.

Parameter	HNSCC Patients (Exosome Isolation) *n* = 21	Control Group (Exosome Isolation) *n* = 10	*p*-Value *	HNSCC Patients (B Cell Isolation) *n* = 23	Control Group (B Cell Isolation) *n* = 23	*p*-Value *
Mean age (range)	64.1 (49–79)	29.2 (20–58)	0.0001	59 (37–74)	56 (27–84)	0.58
Sex						
Female	3 (14.3%)	6 (60%)		9 (39.1%)	13 (56.5%)	
Male	18 (85.7%)	4 (40%)	0.0001	14 (60.9%)	10 (43.5%)	0.24
T classification						
T1	1 (4.8%)			4 (17.4%)		
T2	5 (23.8%)			8 (34.8%)		
T3	6 (28.6%)			5 (21.7%)		
T4	9 (42.9%)			6 (26.1%)		
N classification						
N0	0			7 (30.4%)		
N1	3 (14.3%)			5 (21.7%)		
N2	13 (61.9%)			7 (30.4%)		
N3	5 (23.8%)			4 (17.4%)		
M classification						
M0	21 (100%)			23 (100%)		
M1	0			0		
HPV-status						
Positive (p16)	3 (14.3%)			7		
Localization						
Pharynx	12 (57.1%)			14 (60.9%)		
Larynx	5 (23.8%)			4 (17.4%)		
Mouth	4 (19.0%)			5 (21.7%)		

*: Statistical comparison of age and gender distribution between HNSCC and NC patients.

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
