# Peer review of "Circulating Exosomes Inhibit B Cell Proliferation and Activity"

_cancers, 2020, doi:10.3390/cancers12082110_

Round 1

Reviewer 1 Report

The authors tried to determine how tumor-derived exosome affects B cells in HNSCC patients. These studies could provide important information, particularly for the identification of molecular mechanisms by which HNSCC cancer microenvironment controls cancer progression.

In this study, the exosomes isolated from HNSCC patients and healthy subjects were treated on B cells derived from healthy subjects to confirm the effect on B cell growth and surface receptor expression. However, it is thought that these experimental conditions will be different from the in vivo context, and it is difficult to confirm the effect of exosomes derived from tumor tissues of HNSCC patients on B cells systematically. Therefore, the author should identify the surface expression of checkpoint receptors by separating B cells from HNSCC patients as well as healthy subjects. In addition, B cells isolated from healthy subjects and HNSCC patients should be primarily cultured and then treated with different-origin exosomes to see if there are differences in B cell proliferation and surface receptor expression.

Minor revision

- It is necessary to include basic information (age, gender) on healthy subjects in Table 1 and to perform statistical analysis to see if there are any differences between these groups.

- Pleases increase the font size of Figure 2, A.

Reviewer 2 Report

Overall, the authors report that plasma-derived exosomes inhibit the function of healthy B cells. The results are interesting but a number of issues need to be addressed.

The western blot results are unconvincing. It would be helpful if the authors clarified exactly what the lane marker is (it this a positive control, molecular weight marker?). Traditionally, TSG101 and EpCam are represented by a single band each, not doublets as shown in Figure 1. The images are cropped so it is difficult to ascertain whether there is additional background. According to the MISEV guidelines (that the authors cite), additional markers should be included to verify the purity of the exosomes.

In relation to this data, the methodology for western blotting and TEM are absent. These are crucial and should be included in supplementary materials if the word limit prevents inclusion in the main manuscript. Additional details on the protein quantitation assay are needed (for example, were the exosomes lysed prior to quantitation? If yes, how?).

The authors show that exosomes, regardless of origin, appear to inhibit B cell numbers and survival. This is apparent after 2 days (48 hours) and appears to not change dramatically after this time (at 4 days). This result is interesting and provocative. However, this manuscript would be significantly strengthened if there was more transparency regarding the numbers of replicates used to generate the data. There were 10 controls and 21 HNSCC included in this study but it is unclear if all were used in all experiments.

The authors have used plasma-derived exosomes which will include a heterogeneous mixture of exosome from a range of different cell types that are likely to change from day to day, from person to person in response to factors that we are yet to appreciate. The work presented in this manuscript is important if we are to advance the field but some commentary in consideration of this variability may be helpful.

The authors have also measured the effect of exosomes on immune checkpoint markers on B cells. They show that: exosomes from healthy controls drove an increase in the expression of PD1, CTLA4, LAG3, and CD137; exosomes from HNSCC downregulated expression of GITR; and exosomes, regardless of origin, downregulated expression of BTLA4. Other markers (TIM3, CD27, and OX40) were unchanged. These results are based on 48 hours of co-culturing B cells with exosomes because the variance after 4 days was too great. This raises some concerns about the other data in the manuscript that is reported after 4 days of co-culturing. In the context of exosomes inhibiting B cell survival, the relevance of up- or down-regulation of immune checkpoint markers is unclear. The discussion on these results is far too brief and technical and offers no real interpretation of the findings. Additional uptake studies showing whether exosomes enter the B cells to elicit this effect may be helpful.

The authors also report that CD39 expression was downregulated by exosomes (regardless of origin, healthy control, or HNSCC). Technically, this result is sound but this finding was not discussed in sufficient detail so the relevance and significance are unclear.

Analyses of the B-cell receptor pathway are also unconvincing. The experiment lacks a negative control (eg co-culture of B cells with no exosomes) and there is a large amount of variability in the results. This finding seems too preliminary to include in a publication in its current state.

There is some minor repetition through (for example page 5, lines 141-146 are directly repeating lines 135-140).

Round 2

Reviewer 1 Report

Major revision

Statistical analysis is still needed for the differences in Table 1, age and gender. Because exosome composition might be different depending on age and gender. In particular, it is difficult to understand that the age distribution between the two populations of non-cancer and HNSCC subjects that separated from exosome is too different.

Fig. 3 In (D), (E), statistically analyze the differences between HNSCC and NC

Minor revision
Capitalize all the subtitles of the figure legends.

eg. (a) to (A)

Author Response

To Editor                                                                          Re: cancers-860778

Journal Cancers

Special Issue Advances in Head and Neck Squamous Cell Carcinoma (HNSCC)

Dear Prof. Mok,

Please find enclosed our revised manuscript referred above and entitled ‘Circulating Exosomes inhibit B cell proliferation and activity’. We have revised the manuscript as suggested by the two reviewers. Our revisions on responses to reviewers’ comments are itemized below.

Reviewer 1:

Statistical analysis is still needed for the differences in Table 1, age and gender. Because exosome composition might be different depending on age and gender. In particular, it is difficult to understand that the age distribution between the two populations of non-cancer and HNSCC subjects that separated from exosome is too different.

Statistical analysis for differences in age and gender distribution was performed and p-values inserted into Table 1. Indeed, age and gender distributions are significantly different, which could be expected to confound differences seen between results from co-cultivation experiments with HNSCC and normal controls exosomes. However, in our study, exosomes of both, HNSCC patients and normal controls, inhibited proliferation and viability of B cells. Therefore, we do not expect to see such differences when using exosomes from older healthy donors. There were some differences between HNSCC and NC exosomes concerning their effects on B cell checkpoint receptor expression. However, effects were overall larger for exosomes of healthy donors. To address these important issues, we added the following paragraph to the discussion:

‘Age and gender distribution between NC and HNSCC patients enrolled for exosome isolation were significantly different with NC patients being younger and with a higher female fraction. However, both types of exosomes inhibited proliferation and viability of B cells in a similar way. Therefore, we do not expect to see a difference when enrolling older NC patients. Still, differences seen in checkpoint receptor expression after co-cultivation of B cells with HNSCC versus NC exosomes might be confounded by the different demographic composition of HNSCC and NC patients.’

3 In (D), (E), statistically analyze the differences between HNSCC and NC.

Statistical analysis was performed. There were no significant differences between HNSCC and NC exosomes. p-values for Fig. 3D: 0,6185 (day 2), 0,1429 (day 3), 0,4833 (day 4). p-values for Fig. 3E: 0,0714 (day 2), 0,2858 (day 3), 0,2165 (day 4). The legend was completed as follows: ‘There were no significant differences between HNSCC and NC exosomes’.

Capitalize all the subtitles of the figure legends, e.g. (a) to (A).

Figure subtitles were capitalized as requested.

Reviewer 2 Report

The revisions are comprehensive and have excelled at addressing my comments/concerns.

Author Response

To Editor                                                                          Re: cancers-860778

Journal Cancers

Special Issue Advances in Head and Neck Squamous Cell Carcinoma (HNSCC)

Dear Prof. Mok,

Please find enclosed our revised manuscript referred above and entitled ‘Circulating Exosomes inhibit B cell proliferation and activity’. We have revised the manuscript as suggested by the two reviewers. Our revisions on responses to reviewers’ comments are itemized below.

Reviewer 2:

The revisions are comprehensive and have excelled at addressing my comments/concerns.

We thank the reviewer for his review and comments.

Round 3

Reviewer 1 Report

It is unfortunate in some points, but I think it has been supplemented a lot.